# Shift Work, Shifted Diets: An Observational Follow-Up Study on Diet Quality and Sustainability among Healthcare Workers on Night Shifts

**DOI:** 10.3390/nu16152404

**Published:** 2024-07-24

**Authors:** Semra Navruz-Varlı, Hande Mortaş

**Affiliations:** Department of Nutrition and Dietetics, Faculty of Health Sciences, Gazi University, 06490 Ankara, Türkiye; semranavruz@gazi.edu.tr

**Keywords:** healthcare workers, sustainable nutrition, diet quality, night shift

## Abstract

This study aimed to investigate the change in diet quality in addition to dietary adherence to the planetary health diet during night shifts in healthcare workers. This observational follow-up study involved 450 healthcare workers working night shifts (327 females, 123 males). A survey form requesting sociodemographic information (gender, age, marital status), job title, sleeping duration during the night shift, 24 h dietary records for pre-night-shift, during night shift, and post-night-shift, and anthropometric measurements (body weight and height) was applied. The scores of the Planetary Health Diet Index (PHDI) and the Healthy Eating Index 2020 (HEI-2020) were calculated according to the dietary records. The total HEI-2020 and PHDI scores decreased significantly (*p* < 0.05) during the night shift (44.0 ± 8.8 and 48.3 ± 13.2, respectively) compared to pre-night-shift (46.1 ± 9.2 and 51.9 ± 13.4, respectively) and increased post-night-shift (44.7 ± 9.9 and 50.6 ± 14.9, respectively), with no statistically significant difference between pre- and post-night-shift. There was a significant main effect of night shift working on total PHDI (F(896, 2) = 8.208, *p* < 0.001, η_p_^2^ = 0.018) and HEI-2020 scores (F(894, 2) = 6.277, *p* = 0.002, η_p_^2^ = 0.014). Despite healthcare workers’ knowledge of health factors, night shifts lead to poor dietary choices. To improve diet quality and sustainability, it is crucial to enhance access to healthy food options in their work environment.

## 1. Introduction

Healthcare workers make up one-third of shift workers, while nurses represent the largest group [1]. Shift work schedules are required for individuals to access healthcare services 24 h a day. Almost a quarter of healthcare workers regularly work night shifts [2]. The well-being of healthcare professionals directly affects the health of the patients who need their care. Although practices related to the safety and health of patients come to the fore in healthcare institutions, the safety and health of healthcare professionals should also be supported at the same rate. A natural connection has been reported between the healthcare professional workforce and basic self-care activities (e.g., physical activity, diet quality). In healthcare workers who experience high levels of stress and fatigue, sleep quantity/quality, diet quality, and physical activity cannot reach the desired levels, and this may threaten health [3,4].

Although health professionals have advanced health knowledge, they have difficulty in implementing health-promoting behaviors recommended at the national/international level. The main reasons for this are long working hours and shift work (especially night shift work). These harmful working conditions increase the risk of physiological and psychological disorders. Shift work mainly changes the diet undesirably [5]. When working the night shift, food intake differs from day to night. Night shift workers have a higher meal frequency and lower diet quality than day shift workers [6,7]. As a first step to ensuring the implementation of targeted wellness strategies, it is critical to determine changes in nutrition as well as sleep and physical activity assessments under extended hours and night shift working conditions. The average diet quality of nurses working night shifts is low. Nurses working the night shift need resources to support diet quality to reduce the adverse effects caused by working conditions [8]. Additionally, fatty and salty foods were found to be negatively associated with diet quality in female nurses working night shifts [2]. Attention is drawn to the importance of intervention studies on meal timing and diet quality of healthcare workers working shifts [9].

Global changes in nutrition are associated with intensive production methods that contribute to environmental degradation (greenhouse gas emissions, land use change, land degradation, water pollution, etc.) while causing an increase in non-communicable chronic diseases. The Planetary Health Diet Index (PHDI) was developed to measure compliance with dietary evidence established by the EAT-Lancet Commission. It has been reported that nutritional adequacy in terms of vitamins, fiber, and minerals will be increased by changing diets according to EAT-Lancet recommendations. The need for policy action to support healthier and more-sustainable diets worldwide has been highlighted [10]. It is thought that evaluating the compliance of healthcare professionals, who are pioneers in protecting and improving public health, with the planetary health diet and revealing the current situation will contribute to filling the gap in the literature.

To our knowledge, no study has been found that evaluates the differences in healthcare workers’ diet quality and planetary health diet compliance by following up throughout the periods of pre-night-shift, during, and post-night-shift. Therefore, this study aimed to investigate the change in diet quality and dietary adherence to the planetary health diet during night shifts in healthcare workers.

## 2. Materials and Methods

### 2.1. Participants and Study Design

This study was conducted as a prospective observational follow-up study with the voluntary participation of 450 nurses, doctors, emergency medical technicians, and ambulance care technicians working in private, university, and state hospitals and private health institutions in Ankara, the capital of Turkey. The study was conducted between June 2023 and January 2024. Criteria for inclusion of healthcare professionals in the study were: being between the ages of 20 and 40 years, not having a disease that affects food consumption (situations that limit the consumption of various foods, such as lactose intolerance and celiac), not being a vegetarian or vegan, not being on a weight loss diet, not having any food allergies, not being pregnant or breastfeeding, and working shifts for at least a year. Individuals (n = 22) who started working the day shift instead of the night shift, who did not provide 24 h dietary records, or who lost communication during the study period were excluded from the study.

Healthcare workers’ shift working patterns were 24 h a day, from 08:00 a.m. to 08:00 a.m. the next day. The data obtained in this period were evaluated as “during night shift” in this article. The working schedule of healthcare professionals was 24 h work/24–48 h off. In the article, the day before the 24 h working day was expressed as the “pre-night-shift” (off day). The day after a 24 h working day was defined as the “post-night-shift” (off day). The healthcare workers were followed up for three days: pre-night-shift, during night shift, and post-night-shift.

Ethical approval was obtained from the Ethical Committee of Gazi University (Date: 16 April 2024, No: 2023-851-07). In addition, written informed consent was obtained from the participants in the study. This research was carried out following the Declaration of Helsinki.

### 2.2. Data Collection Tools

In this study, a survey form requesting sociodemographic information (gender, age, marital status), job title, sleeping duration during the night shift, and 24 h dietary records pre-night-shift, during night shift and post-night-shift was applied by the researchers via a face-to-face interview method. The participants’ body weight and height were measured using a BC 532 Innerscan scale (Tanita Corporation, Tokyo, Japan) and stadiometer, respectively. Body mass index (BMI) was calculated as body weight in kilograms divided by the square of the height in meters (kg/m^2^) and categorized [11].

The researchers conducted a nutritional assessment by collecting 24 h dietary records from all participants for three days including “pre-”, “during” and “post-night-shift”. Participants were instructed to log everything they consumed, including foods, beverages, sauces, and condiments. In addition, the mealtimes were recorded, and information was obtained about which meal they consumed that food in. The healthcare workers prepared and brought in the meals they consumed during the night shift from home or obtained them from places such as canteens and cafeterias at work. Individuals self-declared the ingredients of the meals they prepared while keeping 24 h dietary records. However, the ingredients that went into the ready-made meals they purchased from places such as canteens and cafeterias were obtained through the Standard Food Recipes book [12]. The collected data were then analyzed for total energy and nutrient intake using the BeBiS program, version 7.2.

### 2.3. Instruments to Evaluate Diet Quality and Sustainability

The United States Department of Agriculture (USDA) developed the Healthy Eating Index (HEI) in 1995 based on the American Dietary Guidelines [13]. This index was updated in 2005, 2010, and 2015. The components and standards of the HEI-2015 remained unchanged in the HEI-2020. The HEI-2020 retains the same 13 components and scoring standards as the HEI-2015, despite being renamed to reflect its alignment with the 2020–2025 Dietary Guidelines for Americans.

The HEI-2020, utilized in this study, includes 13 components: nine that should be consumed in adequate amounts and four that should be consumed in moderation. The nine desired components are total fruit, whole fruit, total vegetables, green leafy vegetables and fresh legumes, whole grains, dairy products, protein foods, seafood and plant-based proteins, and fatty acids. The four components to be limited are refined grains, sodium, added sugars, and saturated fats. Each component in the index is scored between 0 and 5 or 0 and 10, with low scores indicating poor nutrition and high scores indicating good nutrition [14].

The Planetary Health Diet Index (PHDI) was created by Cacau et al. (2021) based on the dietary recommendations from the EAT-Lancet Commission [15]. The index scores range from 10 to 5 points per component, with a total possible score of 0–150. Sixteen diet components are assessed using food records. The components and their maximum points are: red meat (10 points), nuts and peanuts (10 points), legumes (10 points), chicken and its substitutes (10 points), fish and seafood (10 points), eggs (10 points), fruit (10 points), vegetables (10 points), the ratio of dark green leafy vegetables to other vegetables (5 points), the ratio of red vegetables to other vegetables (5 points), whole grains (10 points), milk and its products (10 points), unsaturated fats (10 points), animal fats (10 points), and added sugars (10 points).

### 2.4. Physical Activity Assessment

Healthcare workers’ physical activity adequacies were evaluated using the Brief Physical Activity Assessment Tool developed by Marshall et al. (2005) [16]. The assessment includes two questions: one about the frequency and duration of vigorous-intensity physical activity, and the other about the frequency and duration of moderate-intensity activities (including walking) performed in a typical week. A scoring algorithm combines the responses to these questions to determine if individuals are classified as “adequately active”.

### 2.5. Statistical Analysis

Continuous variables were presented as the arithmetic mean with standard deviation, while categorical variables were shown as percentages. The HEI-2020 and PHDI total scores were categorized according to descriptive characteristics, including gender, age group, BMI group, marital status, job title, sleeping status during the night shift, and physical activity adequacy, and the changes between pre-, during and post-night-shift were compared using repeated-measures ANOVA (Table 1). If there was a significant difference in the repeated-measures ANOVA test, Tukey’s post-hoc test was used for pairwise comparisons of the parameters. Changes in the HEI-2020 and PHDI total scores and subcomponent scores between pre-, during, and post-night-shift were compared using repeated-measures ANOVA (Table 2). Tukey’s post-hoc test was used for pairwise comparisons of the parameters. The effects of individuals’ night shift changes (pre-, during, and post-) and, the interactions of the night shift changes with gender, age groups, BMI groups, marital status, branch of work, sleeping status during the night shift, and physical activity on the total PHDI and HEI-2020 scores were analyzed using two-way between-subjects ANOVA (Table 3). The results were interpreted with 95% confidence. Statistical analysis was conducted using IBM SPSS Statistics version 28.0.1.0, with significance determined at *p* < 0.05.

## 3. Results

The descriptive characteristics of the participants and the total PHDI and total HEI-2020 scores pre-, during and post-night-shift, categorized according to these characteristics, are shown in Table 1. It has been shown that total PHDI scores in females were lowest during the night shift (48.2 ± 13.8; *p* < 0.001). Total HEI-2020 scores in females were found to be significantly lower during and post-night-shift than pre-night-shift (43.8 ± 8.8; 44.6 ± 10.2; 46.6 ± 9.4, respectively; *p* < 0.001). There was no significant change in total HEI-2020 scores in men according to shift patterns (*p* > 0.05). However, it was determined that total PHDI scores in males decreased significantly during and post-night-shift (48.8 ± 11.7 and 50.8 ± 14.9, respectively) compared to pre-night-shift (51.9 ± 13.3; *p* < 0.001). It has been shown that the total PHDI and total HEI-2020 scores of healthcare workers according to their age groups decreased significantly during the night shift compared to pre-night-shift. Total PHDI and total HEI-2020 scores of individuals decreased during the night shift in normal-body-weight and overweight groups, and there was still a significant difference post-night-shift compared to pre-night-shift, that is, there was no increase in total PHDI and HEI-2020 scores after the shift. Similarly, it was determined that total HEI-2020 and PHDI scores decreased significantly during the night shift in single and married individuals compared to pre-night-shift (*p* < 0.001). When looked at according to the professions of healthcare workers, it has been shown that both total HEI-2020 and total PHDI scores decrease significantly during the night shift in nurses, doctors, and emergency medical technicians, except for ambulance maintenance technicians (*p* < 0.001). A decrease in total HEI-2020 and PHDI scores was detected in individuals who did not sleep during the night shift and those who slept for 2 h during the night shift, while a similar decrease was observed in those who did sufficient physical activity (*p* < 0.001).

The change in the mean and standard deviation values of the HEI-2020 and PHDI subcomponents of healthcare workers pre-, during and post-night-shift is shown in Table 2. It was revealed that the total HEI-2020 scores of individuals decreased significantly during the night shift (44.0 ± 8.8) compared to pre-night-shift (46.1 ± 9.2; *p* = 0.002) and increased post-night-shift (44.7 ± 9.9), with no statistically significant difference between pre- and post-night-shift. The change observed in subcomponent scores during the night shift compared to the pre-night-shift was similarly demonstrated in the sodium (*p* = 0.014), whole fruit (*p* = 0.001), total fruit (*p* < 0.001), and added sugar (*p* = 0.010) subcomponents. It was revealed that the total PHDI scores of individuals decreased significantly during the night shift (48.3 ± 13.2) compared to pre-night-shift (51.9 ± 13.4; *p* < 0.001) and increased post-night-shift (50.6 ± 14.9), with no statistically significant difference between pre- and post-night-shifts. The change observed in subcomponent scores during the night shift compared to the pre-night-shift was similarly demonstrated in the chicken and substitutes (*p* < 0.001), vegetable oils (*p* = 0.001), animal fats (*p* = 0.001), whole cereals (*p* = 0.027), and added sugar (*p* = 0.003) subcomponents. However, it was found that individuals’ legumes (*p* = 0.013), eggs (*p* < 0.001), and dairy (*p* = 0.010) PHDI subcomponent scores increased significantly during the night shift compared to pre-night-shift and decreased again post-night-shift.

The effects of individuals’ night shift changes (pre-, during, and post-night-shift) and the interactions of the night shift changes with gender, age group, BMI group, marital status, branch of work, sleeping status during the night shift, and physical activity on the total PHDI and HEI-2020 scores are shown in Table 3. There was a significant main effect of night shift working on total PHDI (F(896, 2) = 8.208, *p* < 0.001, η_p_^2^ = 0.018) and HEI-2020 scores (F(894, 2) = 6.277, *p* = 0.002, η_p_^2^ = 0.014). In contrast, there was no significant main effect of night-shift-working interactions with gender, age group, BMI group, marital status, branch of work, sleeping status during the night shift, and physical activity on total PHDI and HEI-2020 scores (*p* > 0.05).

The changes in contributions of individuals’ dietary macronutrients to their dietary energy, diet quality, and diet sustainability according to whether they were pre-, during, or post-night-shift are visualized in Figure 1, Figure 2 and Figure 3. In the figures, the night shift sequence is expressed with moonlight while pre- and post-night-shift are expressed with sunlight. The size of the world map symbolizes diet quality. It is stated that the larger the world map (pre-night-shift), the higher the diet quality. Meanwhile, the smallest world map diameter indicates the lowest diet quality (during night shift) and the medium-sized world map (post-night-shift) indicates a higher diet quality than the lowest diet quality. According to the difference among the shift periods, it is symbolized that as the diameter of the world map decreases, the quality of the diet also decreases. The vibrant shades of green color on tree leaves vary depending on their dietary sustainability. The most vibrant shade of green indicates the highest diet sustainability (pre-night-shift), while paler green indicates lower diet sustainability (post-night-shift), and the palest green indicates the lowest diet sustainability (during night shift). Moreover, the proportional distributions of green colors and the world map diameter are shown for illustrative purposes only, not mathematical purposes. No significant change was observed in the contribution of individuals’ dietary protein intake to dietary energy according to whether they were pre-, during, or post-night-shifts. While the contribution of individuals’ dietary carbohydrate intake to dietary energy decreased significantly during the night shift (43.7%; *p* < 0.001) compared to pre-night-shift (44.4%), it increased significantly post-night-shift (45.4%; *p* = 0.047) and reached a level where there was no statistical difference compared with pre-night-shift (*p* = 0.156). While the contribution of individuals’ dietary fat intake to dietary energy increased significantly during the night shift (41.4%; *p* = 0.008) compared to pre-night-shift (40.7%), it decreased post-night-shift (39.9%; *p* < 0.001) and reached a level where there was no statistical difference compared with before the shift (*p* = 0.253).

## 4. Discussion

In this observational follow-up study conducted on healthcare workers working night shifts, it was shown that both the diet sustainability and diet quality of individuals decreased significantly during night shift working compared to before and increased again after the night shift. This change occurred regardless of the healthcare professionals’ job title, age, ability to sleep during shifts, and physical activity status.

It has been shown that the circadian rhythm disruption experienced by healthcare professionals as a result of working night shifts causes many negative health consequences, such as impaired glucose tolerance [17], decreased insulin sensitivity [18,19], obesity [20], cardiometabolic diseases [21], and cancer [22]. It was found that the health problems seen in individuals were not only caused by circadian disruptions as a result of changes in the sleep–wake cycle, but night shifts can also cause diseases by changing eating habits in an unhealthy way [20,23,24,25,26]. In order to intervene in changing eating habits during a night shift, it is necessary first to reveal the changes. In this study, which examined the diet quality and diet sustainability of the night shift in healthcare workers, a group with basic knowledge about health protection, it was shown that there were negative changes in nutritional status during the night shift. Participants’ diet quality measured using HEI-2020 and dietary sustainability measured using PHDI were found to be significantly lower during the night shift compared to pre- and post-night-shift. Similarly, it has been revealed that individuals working night shifts frequently choose unhealthy snacks as a strategy to stay awake during their shifts, and these snacks are often high in fat, sodium, and added sugar [23,27,28]. In this study, it was shown that the index scores of individuals according to their saturated fatty acid intake were low pre-, during, and post-night-shift, without any significant differences among them. Additionally, individuals’ diet quality scores from added sugar intake were shown to significantly decrease during night shifts (4.9 ± 4.6, 4.1 ± 4.5, 4.6 ± 4.7, pre-, during, and post-night-shift, respectively; *p* = 0.010), supporting previous studies.

In addition, when it was investigated as to how the effects of a night shift on the decrease in diet quality would change when combined with these sociodemographic data, it was shown that the night shift did not show its effects on diet quality by interacting with these sociodemographic descriptors, and that the night shift was the main influencing factor (F(894, 2) = 6.277, *p* = 0.002, η_p_^2^ = 0.014, Table 3). Similar to this study, it has been revealed that food intake or eating habits change in an “unhealthy” direction [20,24,25,26,27,28,29,30,31] along the same lines as in this study conducted in healthcare workers. However, studies investigating diet quality in night shift workers are quite limited [8]. The evaluation of the HEI-2020 score and its subcomponents, which allow for comparison of the nutritional status of individuals with more objective criteria, has contributed an essential finding to the literature.

Sustainable nutrition, the importance of which has become more evident with the rapid depletion of natural resources, will be an inevitable strategy of this age, not only because of its low environmental impacts and protection of the ecosystem and biodiversity but also because it is a nutritionally adequate and healthy nutrition model [32]. This should also be recommended for night shift workers. In this way, the health of these individuals in the at-risk group working the night shift will be protected while natural resources will also be protected. Sustainable nutrition models, which will become an inevitable proposal of the age by consensus, are characterized by prominently featuring substantial quantities of plant-based foods, including vegetables, fruits, seeds, nuts, legumes, and whole grains, while incorporating only moderate-to-minimal amounts of animal-based products, such as meat, poultry, seafood, eggs, and dairy [15,32,33,34,35]. Considering the foods included in sustainable nutrition recommendations, it appears that they meet the dietary approaches recommended to improve health in individuals working night shifts. Therefore, the implementation of sustainable nutrition approaches in individuals working night shifts reduces the intake of saturated fatty acids and added sugars, which are included in unhealthy eating tendencies in this group [20,23,24,25,26], while increasing the intake of fiber, unsaturated fatty acids, fruits, and vegetables. The supporting data in this study demonstrate that sustainable diet quality decreased when healthcare workers switched to night shifts. During the night shift, healthcare workers’ sustainable diet quality scores of red meat (0.6 ± 2.3), chicken and substitutes (6.6 ± 4.7), and animal fats (4.6 ± 0.4) decreased significantly (*p* < 0.05) compared to pre-night-shift (1.5 ± 3.5, 7.7 ± 4.2, 5.1 ± 0.9, respectively), meaning their consumption of these foods increased significantly. The significant decrease in the consumption of these foods post-night-shift (*p* < 0.05), that is, the increase in the PHDI subscores (2.3 ± 4.1, 8.0 ± 3.9, 5.7 ± 0.9, respectively), strengthens the hypothesis that the changes occur only during the night shift (Table 2). Similarly, studies have shown that meat consumption increases during the night shift, and that individuals tend to eat meals containing high saturated fat, sodium, red meat, and chicken, and especially “fast” foods [29,36,37].

Studies have shown that females tend to have a higher sustainable diet quality than males and are more prone to plant-based foods [38,39,40]. For these reasons, in this study, the sustainable diet quality of females increased faster than males’ post-night-shift. It is thought that the effect of the increase in sustainable diet quality after the night shift depending on the sleeping state during the night shift is also related to the effect of sleep duration on the feeling of hunger. In individuals with less sleep time, the feeling of hunger increases, and as a result, the tendency to foods with poorer nutritional value including “fast” food increases. Studies supporting this interpretation have shown that the feeling of hunger, which causes unhealthy eating choices, is affected by sleep duration [8,41,42]. It has been shown that decreasing sleep duration affects hunger and satiety levels through the mechanism of changing plasma ghrelin and leptin levels, which are hormones that regulate appetite [8,41,42]. The increase in sustainable diet quality after a night shift in individuals with a normal body weight and adequate physical activity level has been interpreted as being related to the fact that the diet choices of individuals who make healthy lifestyle choices may also be healthy. However, when it was investigated as to how the effects of the night shift on the decrease in sustainable diet quality (total PHDI score) would change when combined with these sociodemographic data, it was shown that the night shift did not show its effects on sustainable diet quality by interacting with these sociodemographic descriptors, and that the night shift was the main influencing factor on sustainable diet quality (F(894, 2) = 6.277, *p* = 0.002, η_p_^2^ = 0.014, Table 3).

One of the strengths of this study is that it evaluated night shift workers’ nutritional status by following a large sample before, during, and after the night shift. In addition, this study, in which the diet quality of the participants was calculated according to HEI-2020, has another strength due to the limited number of studies in the literature that have observed changes in diet quality by following night shift workers. Another of its strengths is that it is the first study to investigate the implementation of sustainable nutrition, which will inevitably become a healthy nutrition strategy for every individual due to today’s conditions, including night shift workers. The first limitation of this study is that since the study was conducted only in healthcare workers, the results cannot be generalized to all night shift workers. Secondly, there are limitations in generalizing the findings of the study because the factors affecting individuals’ eating habits and food preferences vary across countries and even geographical regions within the same country. Thirdly, considering that the stress experienced by healthcare workers during night shifts may affect their eating behavior, it will be important to include this parameter in future studies. Lastly, healthcare workers may encounter many emergencies during the night shift. Experiencing these emergencies may also affect individuals’ food consumption. However, in this study, no evaluation was made regarding the emergency situations encountered by healthcare workers working on night shifts, and it is recommended that future studies include this evaluation and exclude from the study periods when unusual emergencies (long surgeries, etc.) occur.

## 5. Conclusions

Compared to before the night shift, both the diet quality and diet sustainability of healthcare workers decreased during the night shift, without any differences according to gender, age group, BMI group, marital status, job title, and napping during the night shift. The significant increase in diet quality and diet sustainability scores again after the night shift supports the finding that the night shift is an important factor affecting these two parameters. It is thought that sustainable nutrition, which is a solution-oriented proposal of the age with its health benefits and strengths such as the protection of natural resources and low environmental impact, will be inevitable to be implemented day by day in health professionals working night shifts.

The fact that these negative dietary changes occur during the night shift, even among healthcare workers who have basic knowledge about the factors affecting the protection and promotion of health, reveals that further steps are needed than just providing information on the subject. The first of these should be to increase the chance of accessing healthy alternatives by evaluating the environment in which healthcare workers access food sources. Other suggestions include organizing break times at work by encouraging healthy eating and ensuring that workplace cafeterias and vending machines contain healthy alternatives. Ensuring that not only healthcare professionals but also managers are informed about sustainable nutrition will make it easier to accelerate the steps to be taken. Finally, it is also valuable to evaluate the nutritional knowledge levels of healthcare professionals in future studies and correlate them with diet quality and sustainable nutritional quality, and to develop suggestions for updates to be made in the undergraduate education curriculum of healthcare professionals based on the results of the research.

## Figures and Tables

**Figure 1 nutrients-16-02404-f001:**
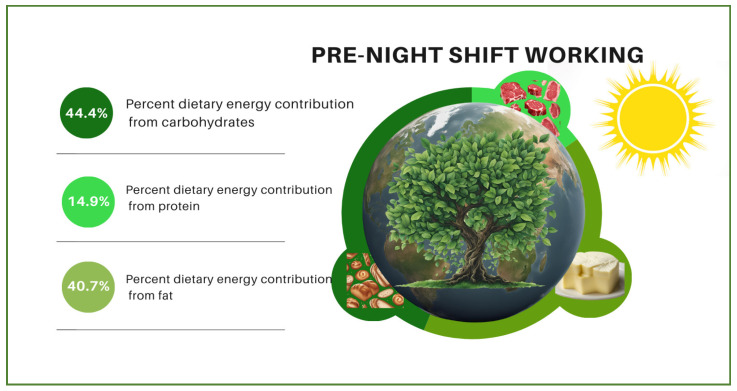
The contributions of individuals’ dietary macronutrients to their dietary energy, diet quality, and diet sustainability pre-night-shift.

**Figure 2 nutrients-16-02404-f002:**
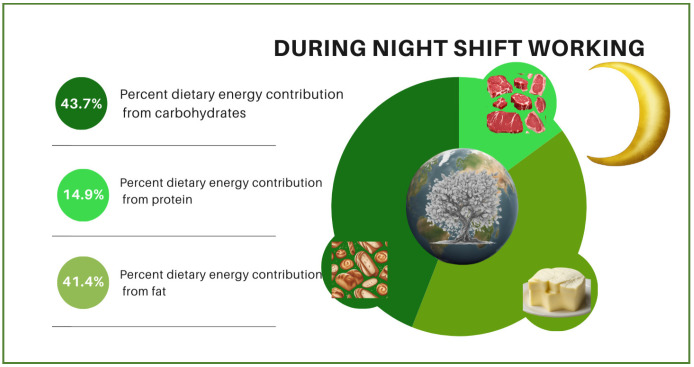
The contributions of individuals’ dietary macronutrients to their dietary energy, diet quality, and diet sustainability while working the night shift.

**Figure 3 nutrients-16-02404-f003:**
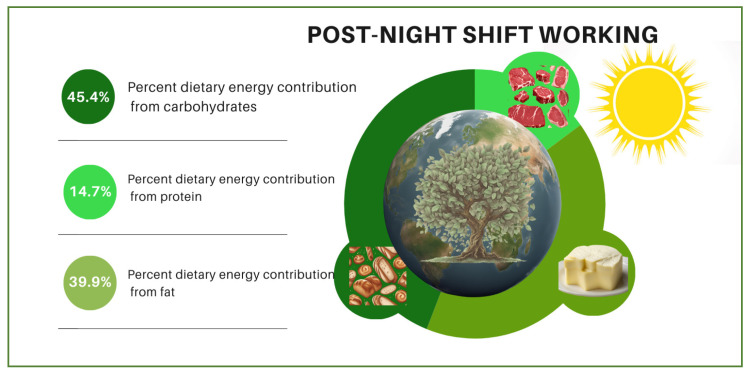
The contributions of individuals’ dietary macronutrients to their dietary energy, diet quality, and diet sustainability post-night-shift.

**Table 1 nutrients-16-02404-t001:** Descriptive characteristics of the health workers.

Variables	Percentages n (%)	Total PHDI Score (Mean ± SD)			Total HEI-2020 Score (Mean ± SD)		
		Pre-Night-Shift	During Night Shift	Post-Night-Shift	Pre-Night-Shift	During Night Shift	Post-Night-Shift
Gender							
Female	327 (72.7)	51.9 ± 13.4 ^a^	48.2 ± 13.8 ^b^	50.8 ± 14.9 ^a^	46.6 ± 9.4 ^a^	43.8 ± 8.8 ^b^	44.6 ± 10.2 ^b^
		F = 10,730.176; *p* < 0.001; η^2^ = 0.971			F = 8034.557; *p* < 0.001; η^2^ = 0.985		
Male	123 (27.3)	51.9 ± 13.3 ^a^	48.8 ± 11.7 ^b^	50.8 ± 14.9 ^a,b^	44.7 ± 8.4	44.6 ± 8.6	45.0 ± 8.9
		F = 5049.914; *p* < 0.001; η^2^ = 0.976			F = 18,506.432; *p* = 0.170; η^2^ = 0.014		
Age group (years)							
20–25	122 (27.1)	49.3 ± 14.1 ^a^	45.5 ± 12.9 ^b^	46.2 ± 14.5 ^a,b^	45.6 ± 9.6 ^a^	43.4 ± 8.4 ^b^	42.7 ± 9.9 ^b^
		F = 3830.169; *p* < 0.001; η^2^ = 0.969			F = 6830.632; *p* < 0.001; η^2^ = 0.983		
26–30	188 (41.8)	52.5 ± 13.2 ^a^	49.7 ± 13.1 ^b^	51.7 ± 14.9 ^a,b^	45.9 ± 9.1 ^a,b^	44.4 ± 8.6 ^b^	46.1 ± 9.9 ^a^
		F = 6845.947; *p* < 0.001; η^2^ = 0.973			F = 10,859.282; *p* < 0.001; η^2^ = 0.983		
31–40	140 (31.1)	53.4 ± 12.8 ^a^	48.9 ± 13.5 ^b^	52.9 ± 14.5 ^a^	46.7 ± 8.8 ^a^	44.1 ± 9.4 ^b^	44.7 ± 9.6 ^b^
		F = 5937.575; *p* < 0.001; η^2^ = 0.977			F = 9026.722; *p* < 0.001; η^2^ = 0.985		
BMI (kg/m^2^)							
Normal weight	288 (64.0)	51.5 ± 13.3 ^a^	48.4 ± 13.3 ^b^	51.2 ± 15.4 ^a^	46.3 ± 9.5 ^a^	44.5 ± 8.5 ^b^	45.2 ± 9.9 ^a^
		F = 4.642; *p* = 0.010; η^2^ = 0.016			F = 3.407; *p* = 0.034; η^2^ = 0.012		
Overweight	135 (30.0)	51.7 ± 13.2 ^a^	47.8 ± 13.5 ^b^	49.4 ± 13.9 ^b^	45.1 ± 8.4 ^a^	43.4 ± 9.2 ^b^	43.7 ± 9.8 ^b^
		F = 3.057; *p* = 0.049; η^2^ = 0.022			F = 1.549; *p* = 0.011; η^2^ = 0.011		
Obese	27 (6.0)	56.8 ± 13.7	50.5 ± 11.6	50.4 ± 13.7	48.8 ± 8.8	43.8 ± 9.3	45.5 ± 10.1
		F = 2.631; *p* = 0.082; η^2^ = 0.092			F = 2.147; *p* = 0.127; η^2^ = 0.076		
Marital status							
Married	200 (44.4)	53.1 ± 12.8 ^a^	49.4 ± 12.9 ^b^	51.7 ± 14.6 ^a,b^	46.7 ± 8.6 ^a^	44.6 ± 9.2 ^b^	45.6 ± 9.7 ^a,b^
		F = 8042.590; *p* < 0.001; η^2^ = 0.976			F = 13,786.891; *p* < 0.001; η^2^ = 0.986		
Single	250 (55.6)	50.9 ± 13.8 ^a^	47.5 ± 13.5 ^b^	49.7 ± 15.1 ^b^	45.6 ± 9.6 ^a^	43.6 ± 8.4 ^b^	44.1 ± 9.9 ^a,b^
		F = 7959.767; *p* < 0.001; η^2^ = 0.970			F = 13,266.537; *p* < 0.001; η^2^ = 0.982		
Job title							
Nurse	267 (59.3)	50.2 ± 13.5 ^a^	46.6 ± 12.9 ^b^	49.1 ± 15.6 ^b^	46.5 ± 9.0 ^a^	44.7 ± 8.6 ^b^	44.6 ± 10.3 ^b^
		F = 8425.803; *p* < 0.001; η^2^ = 0.969			F = 14,954.402; *p* < 0.001; η^2^ = 0.983		
Doctor	36 (8.0)	50.5 ± 13.2 ^a^	46.3 ± 12.9 ^b^	52.6 ± 14 ^a^	45.6 ± 9.6 ^a^	44.9 ± 7.7 ^b^	48.9 ± 9.6 ^a^
		F = 1694.439; *p* < 0.001; η^2^ = 0.980			F = 2088.182; *p* < 0.001; η^2^ = 0.984		
Emergency medical technician	100 (22.2)	54.5 ± 12.8 ^a^	49.7 ± 12.4 ^b^	51.6 ± 12.4 ^a,b^	45.4 ± 9.4 ^a^	42.1 ± 8.4 ^b^	44.1 ± 9.4 ^a,b^
		F = 4620.098; *p* < 0.001; η^2^ = 0.979			F = 7232.001; *p* < 0.001; η^2^ = 0.987		
Ambulance care technician	47 (10.4)	57.0 ± 12.1	56.5 ± 14.2	55.6 ± 14.7	45.7 ± 9.2	43.8 ± 10.5	43.9 ± 7.9
		F = 2685.913; *p* = 0.881; η^2^ = 0.003			F = 2789.670; *p* = 0.489; η^2^ = 0.015		
Sleeping during night shift							
Not sleeping	118 (26.2)	47.1 ± 11.9 ^a^	43.9 ± 13.4 ^b^	46.3 ± 13.3 ^a,b^	44.9 ± 8.6 ^a^	42.8 ± 7.6 ^b^	43.4 ± 9.3 ^a,b^
		F = 4145.019; *p* < 0.001; η^2^ = 0.973			F = 8357.545; *p* < 0.001; η^2^ = 0.986		
0–2 h sleep	332 (73.8)	53.6 ± 13.4 ^a^	49.9 ± 13.4 ^b^	52.1 ± 15.1 ^a^	46.5 ± 9.3 ^a^	44.5 ± 9.1 ^b^	45.2 ± 10.1 ^a,b^
		F = 13,167.330; *p* < 0.001; η^2^ = 0.975			F = 18,804.628; *p* < 0.001; η^2^ = 0.983		
Physical activity status							
Inadequate	170 (37.8)	48.9 ± 13.1 ^a^	45.8 ± 12.5 ^b^	48.1 ± 15.3 ^a,b^	46.7 ± 8.8	45.7 ± 8.4	45.1 ± 9.9
		F = 10,383.005; *p* < 0.001; η^2^ = 0.974			F = 9883.626; *p* = 0.189; η^2^ = 0.010		
Adequate	280 (62.2)	53.7 ± 13.2 ^a^	49.9 ± 13.5 ^b^	52.1 ± 14.4 ^a^	45.7 ± 9.3 ^a^	43.0 ± 8.9 ^b^	44.5 ± 9.9 ^a,b^
		F = 6230.596; *p* < 0.001; η^2^ = 0.974			F = 16,912.648; *p* < 0.001; η^2^ = 0.984		

^a,b^ represent the statistically significant differences among the line groups at *p* < 0.05. BMI: body mass index; HEI: Healthy Eating Index; SD: standard deviation; PHDI: Planetary Health Diet Index.

**Table 2 nutrients-16-02404-t002:** Diet quality and sustainable nutrition trends of health workers pre-, during and post-night-shifts according to the HEI-2020 and PHDI.

Indices and Their Components	$\mathrm{Pre}-\mathrm{Night}-\mathrm{Shift}\;(\bar{x}\pm \;\mathrm{SD})$	$\mathrm{During}\;\mathrm{Night}\;\mathrm{Shift}\;\;(\bar{x}\pm \;\mathrm{SD})$	$\mathrm{Post}-\mathrm{Night}-\mathrm{Shift}\;\;(\bar{x}\pm \;\mathrm{SD})$	F	*p*	η^2^
HEI-2020 score	46.1 ± 9.2 ^a^	44.0 ± 8.8 ^b^	44.7 ± 9.9 ^a^	6.277	0.002	0.014
HEI-2020 components scores [$\bar{x}\;\pm \;\mathrm{SD}$]						
Whole grains	0.8 ± 2.5 ^a^	0.5 ± 1.9 ^b^	0.7 ± 2.3 ^b^	3.633	0.027	0.008
Refined grains	7.6 ± 2.9	7.2 ± 2.9	7.3 ± 3.3	2.676	0.069	0.006
Seafood and plant proteins	4.9 ± 0.7 ^a^	4.9 ± 0.5 ^a^	4.8 ± 0.9 ^b^	5.626	0.006	0.012
Sodium	9.4 ± 1.9 ^a^	6.7 ± 1.3 ^b^	9.4 ± 1.9 ^a^	4.428	0.014	0.010
Dairy	3.2 ± 2.1	3.1 ± 2.0	3.4 ± 2.7	1.895	0.153	0.004
Greens and beans	3.5 ± 1.7 ^a^	3.5 ± 1.6 ^a^	3.2 ± 1.8 ^b^	5.796	0.003	0.013
Total vegetable	3.2 ± 1.5	3.2 ± 1.4	3.1 ± 1.6	0.348	0.706	0.001
Whole fruit	3.5 ± 1.9 ^a^	2.1 ± 1.7 ^b^	2.4 ± 2.0 ^a^	6.568	0.001	0.014
Total fruit	1.7 ± 1.6 ^a^	1.3 ± 1.2 ^b^	1.7 ± 1.7 ^a^	10.502	<0.001	0.023
Added sugar	4.9 ± 4.6 ^a^	4.1 ± 4.5 ^b^	4.6 ± 4.7 ^a^	4.662	0.010	0.010
MUFA/PUFA ratio	0.1 ± 0.6 ^b^	0.1 ± 0.5 ^b^	0.2 ± 1.3 ^a^	5.001	0.014	0.011
Saturated fatty acids	0.0 ± 0.0	0.0 ± 0.0	0.1 ± 0.4	1.000	0.318	0.002
PHDI score [$\bar{x}\;\pm \;\mathrm{SD}$]	51.9 ± 13.4 ^a^	48.3 ± 13.2 ^b^	50.6 ± 14.9 ^a^	8.208	<0.001	0.018
PHDI components scores [$\bar{x}\;\pm \;\mathrm{SD}$]						
Red meat	1.5 ± 3.5 ^a^	0.6 ± 2.3 ^b^	2.3 ± 4.1 ^c^	27.081	<0.001	0.057
Nuts and peanuts	5.3 ± 4.2 ^a^	5.6 ± 4.2 ^a^	4.7 ± 4.4 ^b^	5.163	0.006	0.011
Legumes	3.2 ± 3.8 ^b^	3.5 ± 3.8 ^a^	2.7 ± 3.8 ^b^	4.332	0.013	0.010
Chicken and substitutes	7.7 ± 4.2 ^a^	6.6 ± 4.7 ^b^	8.0 ± 3.9 ^a^	13.943	<0.001	0.030
Fish and seafood	0.0 ± 0.0	0.0 ± 0.0	0.1 ± 0.1	0.640	0.491	0.001
Eggs	0.4 ± 1.5 ^a^	0.9 ± 2.4 ^b^	2.9 ± 3.7 ^a^	106.921	<0.001	0.192
Fruits	6.9 ± 3.7	6.4 ± 3.6	6.4 ± 3.9	2.4560	0.086	0.005
Vegetables	6.7 ± 2.5 ^a^	8.7 ± 2.5 ^a^	8.3 ± 3.0 ^b^	3.469	0.033	0.008
DGV/total ratio	0.5 ± 0.3	0.4 ± 0.3	0.4 ± 0.3	0.056	0.944	0.000
ReV/total ratio	0.8 ± 0.3 ^a^	0.9 ± 0.3 ^b^	0.8 ± 0.4 ^b^	6.928	0.001	0.015
Whole cereals	0.7 ± 1.8 ^a^	0.4 ± 0.3 ^b^	0.6 ± 0.3 ^a,b^	3.633	0.027	0.008
Tubers and potatoes	0.3 ± 0.1	0.4 ± 0.1	0.5 ± 0.8	1.907	0.150	0.004
Dairy	2.7 ± 0.3 ^a^	2.9 ± 0.4 ^a^	2.3 ± 0.3 ^b^	4.613	0.010	0.010
Vegetable oils	3.9 ± 0.6 ^a^	3.2 ± 0.7 ^b^	3.1 ± 0.4 ^b^	6.853	0.001	0.015
Animal fats	5.1 ± 0.9 ^a^	4.6 ± 0.4 ^b^	5.7 ± 0.9 ^c^	7.822	0.001	0.017
Added sugar	4.3 ± 0.6 ^a^	3.4 ± 0.4 ^b^	4.1 ± 0.6 ^a^	5.706	0.003	0.013

^a,b,c^ represent the statistically significant differences among the line groups at *p* < 0.05. DGV: dark green vegetables; HEI: Healthy Eating Index; MUFA: mono unsaturated fatty acids; SD: standard deviation; PHDI: Planetary Health Diet Index; PUFA: polyunsaturated fatty acids; ReV: red vegetables.

**Table 3 nutrients-16-02404-t003:** Healthy Eating Index 2020 and Planetary Health Diet Index total score differences according to interactions of the night shift working with descriptive variables.

Variables	SS	*df*	MS	F	*p*	η_p_^2^
PHDI total scores						
Night shift working	2933.499	2	1466.750	8.208	<0.001	0.018
Night shift working * Gender	16.190	2	8.095	0.045	0.956	0.000
Night shift working * Age group	513.499	4	128.375	0.717	0.580	0.003
Night shift working * BMI group	581.021	4	145.255	0.812	0.517	0.004
Night shift working * Marital status	5.662	2	2.831	0.016	0.984	0.000
Night shift working * Job title	777.763	6	129.627	0.724	0.630	0.005
Night shift working * Sleeping	20.722	2	10.361	0.058	0.944	0.000
Night shift working * Physical activity	33.696	2	16.848	0.094	0.910	0.000
HEI-2020 total scores						
Night shift working	975.849	2	487.924	6.277	0.002	0.014
Night shift working * Gender	342.720	2	171.360	2.210	0.110	0.005
Night shift working * Age group	425.992	4	106.498	1.372	0.242	0.006
Night shift working * BMI group	142.162	4	35.541	0.456	0.768	0.002
Night shift working * Marital status	14.640	2	7.320	0.094	0.910	0.000
Night shift working * Job title	604.204	6	100.701	1.298	0.255	0.009
Night shift working * Sleeping	6.188	2	3.094	0.040	0.961	0.000
Night shift working * Physical activity	236.282	2	118.141	1.522	0.219	0.003

BMI: body mass index; HEI: Healthy Eating Index; MS: mean square; SS: sum of squares; PHDI: Planetary Health Diet Index. The * symbol shows the combined interactions of the night shift changes with gender, age group, BMI group, marital status, branch of work, sleeping statues, respectively.

## Data Availability

The data presented in this study are available on request from the corresponding author (handeyilmaz@gazi.edu.tr) due to fact that it is intended to be shared only with researchers working in this field.

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
