# Peer review of "Shift Work, Shifted Diets: An Observational Follow-Up Study on Diet Quality and Sustainability among Healthcare Workers on Night Shifts"

_nutrients, 2024, doi:10.3390/nu16152404_

Round 1

Reviewer 1 Report

Comments and Suggestions for Authors

The study attempts to define the possible changes in diet quality and subjects working night shifts. Diet quality was assessed by validated questionnaires and the results were compared with current guidelines measuring diet sustainability. Diet was measured during three consecutive days from 8:00 AM to 8:00 AM on the next day: a) day pre night shift; b) day of night shift; c) day following nigh shift. In general, during night shift, the quality of dietary intake deteriorated, and marginally improved on the following day, compared to the pre-night-shift. Data were largely independent of socio-demographic variables. I have several reservations, only partly addressed as limitations in the manuscript:

1.     Night-shift causes important changes in the dynamic of meal consumption. While working during the night, it is largely possible that dinner is consumed at home before going to hospital, much earlier compared with common days. Similarly, following night shift, breakfast and lunch might be totally displaced owing to the need of day sleep. As such, the day of pre-night shift may be considered a normal day, whereas the other two days are definitely different. This is correctly identified by questionnaires.

2.     In my hospital experience, nurses and other healthcare personnel frequently indulge in food during night at work. These intakes may be difficult to measure and are likely to be extremely variable, depending on the possible emergencies occurring at night. Was any attempt carried out to measure possible differences in relation to emergencies?

3.     Breakfast in the post-night shift day is frequently anticipated. Was it correctly included in the post-night shift calculation?

4.     Figures: they are difficult to interpret. The dimensions of the plates do not correspond to the modest changes in dietary supply.

5.     Discussion: It is over too long and largely repetitive of the results. It might be halved without any loss of information. As an example, lines from 312 to 332 do not make any comparison with data from the literature or provide new hypothesis. This is also the case of lines from 376 to 386.

Author Response

Response to the Reviewer 1

Thank you very much for your contributions and corrections. Also, thank you very much for your time to review the manuscript in detail. We carefully considered your letter. The revisions in the submitted manuscript are explained as listed below.

Questions for General Evaluation

Q1- The corrections suggested by the reviewers were made and highlighted in red within the article. The introduction has been improved.

Q2- The corrections suggested by the reviewers were made and highlighted in red within the article. The research design has been made more understandable.

Q3- The corrections suggested by the reviewers were made and highlighted in red within the article. The method has been made more understandable.

Q4- The corrections suggested by the reviewers were made and highlighted in red within the article. The results have been made more understandable.

Q5- Thank you very much.

Response to Comments and Suggestions for Authors

Reviewer’s comment: 1. Night-shift causes important changes in the dynamic of meal consumption. While working during the night, it is largely possible that dinner is consumed at home before going to hospital, much earlier compared with common days. Similarly, following night shift, breakfast and lunch might be totally displaced owing to the need of day sleep. As such, the day of pre-night shift may be considered a normal day, whereas the other two days are definitely different. This is correctly identified by questionnaires.

Response: 1. Thank you very much for this visionary comment. As you said, the diet changes completely in shift workers. Although this study evaluated individuals' nutritional content rather than their nutritional timing, it is also important to emphasize the change in meal timing and number of meals, as you said. Thank you very much for your contribution. In the introduction part of the article, attention was also drawn to the changes in meal timing and number of meals in shift workers.

Reviewer’s comment: 2. In my hospital experience, nurses and other healthcare personnel frequently indulge in food during night at work. These intakes may be difficult to measure and are likely to be extremely variable, depending on the possible emergencies occurring at night. Was any attempt carried out to measure possible differences in relation to emergencies?

Response: 2. Thank you very much for sharing this valuable observation. Food consumption of individuals working night shifts often occurs at night as you said. In the 24h food consumption recording method, which is the method we use to evaluate the food consumption of healthcare workers during the night shift, individuals were explained how to fill out the registration form to obtain detailed information about both the portion, time and content of the meal while consuming food. In addition, the ingredients of the ready-made meals provided by individuals were obtained through the Standard Food Recipes book, developed by academics working in this field. However, no evaluation has been made regarding the emergencies experienced by healthcare workers during the night shift. Thank you for your contribution. We added your suggestion on this subject to the limitations of the study section. This information has been added to the limitations and written in red.

Reviewer’s comment: 3. Breakfast in the post-night shift day is frequently anticipated. Was it correctly included in the post-night shift calculation?

Response: 3. As you said, consuming breakfast after a shift is common. While filling out food consumption record forms according to days, mealtimes were also recorded and each evaluation period (pre-, during, and post-night shift working) was evaluated according to the meals consumed within itself. An explanation for this has been added to the method section. Thank you for your contribution.

Reviewer’s comment: 4. Figures: they are difficult to interpret. The dimensions of the plates do not correspond to the modest changes in dietary supply.

Response: 4. Thank you very much for your valuable contribution and for giving us the opportunity to explain this issue in detail. As you said, we made the figures to visualize our results and present the change in the night shift to the reader at first glance. The proportional distribution of green colors and the diameter information on the map were used to display the data visually rather than mathematically. Explanation regarding this has been added to the figure comments.

Reviewer’s comment: 5. Discussion: It is over too long and largely repetitive of the results. It might be halved without any loss of information. As an example, lines from 312 to 332 do not make any comparison with data from the literature or provide new hypothesis. This is also the case of lines from 376 to 386.

Response: 5. Thank you very much for your valuable contribution. We attach great importance to the simple and understandable reception of the article by the reader. As you said, we removed the parts you wrote from the discussion section to make it a little more concise.

Reviewer 2 Report

Comments and Suggestions for Authors

ID: nutrients-3106620

Shift work, shifted diets: An observational follow-up study on diet quality and sustainability among healthcare workers on night shifts. by Navruz-Varlı, et al.

To the Authors:

General comments:

The authors investigated about the change in diet quality and diet sustainability during the night shifts in healthcare workers.  It was shown that both diet quality and diet sustainability of healthcare workers decreased during the night shift compared to pre- and post-night shift.  The theme of this article is interesting, and it is well-structured; however, some points should be addressed to improve the manuscript.

Specific comments:

1. In line 76, the authors described that one of the inclusion criteria is “not having a disease that affects food consumption”.  The authors should clarify what kinds of diseases are included in this situation.

2. It is possible that stress caused by the night shift can affect their eating behavior.  Is the relationship between the stress parameter and HEI-2020 or PHDI already considered?

3. To more clearly elucidate the correlation between night shift and dietary choices, it should also be discussed where the workers get the food during the night shift. Did they cook by themselves or buy at stores?

4. The authors expressed the healthcare workers as having enough knowledge about health protection.  However, the amount of knowledge about healthy eating behavior can vary from person to person even among the healthcare workers.  Did the authors perform the test about the knowledge about nutrition? 

Author Response

Response to the Reviewer 2

Thank you very much for your contributions and corrections. Also, thank you very much for your time to review the manuscript in detail. We carefully considered your letter. The revisions in the submitted manuscript are explained as listed below.

Questions for General Evaluation

Q1- The corrections suggested by the reviewers were made and highlighted in red within the article. The introduction has been improved.

Q2- Thank you very much.

Q3- The corrections suggested by the reviewers were made and highlighted in red within the article. The method has been made more understandable.

Q4- Thank you very much.

Q5- The corrections suggested by the reviewers were made and highlighted in red within the article. The conclusion has been made more understandable.

Response to Comments and Suggestions for Authors

Reviewer’s comment: 1. In line 76, the authors described that one of the inclusion criteria is “not having a disease that affects food consumption”.  The authors should clarify what kinds of diseases are included in this situation.

Response: 1. Thank you very much for your valuable contribution. What is meant here is individuals with diseases such as celiac disease and lactose intolerance that limit the consumption of certain foods. This explanation has been added to the method section.

Reviewer’s comment: 2. It is possible that stress caused by the night shift can affect their eating behavior.  Is the relationship between the stress parameter and HEI-2020 or PHDI already considered?

Response: 2. We agree with your comment. Within the scope of this study, the reflection of stress on eating behavior was not evaluated. We added that this issue is a limitation of this study in the limitations section. It will be important to evaluate this parameter in future studies. We also have new studies planned on the subject. We will consider adding this parameter to those studies. Thank you very much.

Reviewer’s comment: 3. To more clearly elucidate the correlation between night shift and dietary choices, it should also be discussed where the workers get the food during the night shift. Did they cook by themselves or buy at stores?

Response: 3. Thank you very much for giving us the chance to present a nice detail about the method that we needed to write about. Healthcare workers prepare and bring the meals they consume during the night shift from home or obtain them from places such as canteens and cafeterias at work. Individuals self-declared the ingredients of the meals they prepared while keeping food consumption records. However, the ingredients that go into the ready-made meals they buy from places such as canteens and cafeterias were obtained through the Standard Food Recipes book, developed by academics working in this field.

Reviewer’s comment: 4. The authors expressed the healthcare workers as having enough knowledge about health protection.  However, the amount of knowledge about healthy eating behavior can vary from person to person even among the healthcare workers.  Did the authors perform the test about the knowledge about nutrition?

Response: 4. Thank you for your contribution. We find the issue you mentioned very valuable. We did not make such an evaluation within the scope of this study. However, in order to make this contribution visible, we added to the recommendations section that it is important to evaluate the nutritional knowledge levels of healthcare professionals and correlate them with diet quality and sustainable nutritional status in future studies.

Round 2

Reviewer 2 Report

Comments and Suggestions for Authors

ID: nutrients-3106620

Shift work, shifted diets: An observational follow-up study on diet quality and sustainability among healthcare workers on night shifts. by Navruz-Varlı, et al.

To the Authors:

The authors responded to my comments and added information in the revised manuscript.  The contents of the article have become much clearer and more constructive.